# DEEP ENSEMBLES WITH HIERARCHICAL DIVERSITY PRUNING

## ABSTRACT

Diverse deep ensembles hold the potential for improving accuracy and robustness of deep learning models. Both pairwise and non-pairwise ensemble diversity metrics have been proposed over the past two decades. However, it is also challenging to find the right metrics that can effectively prune those deep ensembles with insufficient ensemble diversity, thus failing to deliver effective ensemble accuracy. In this paper, we first compare six popular diversity metrics in the literature, coined as Q metrics, including both pairwise and non-pairwise representatives. We analyze their inherent limitations in capturing the negative correlation of ensemble member models, and thus inefficient in identifying and pruning low quality ensembles. We next present six HQ ensemble diversity metrics by extending the existing Q-metrics with three novel optimizations: (1) We introduce the concept of focal model and separately measure the ensemble diversity among the deep ensembles of the same team size with the concept of focal model, aiming to better capture the negative correlations of member models of an ensemble. (2) We introduce six HQ-diversity metrics to optimize the corresponding Q-metrics respectively in terms of measuring negative correlation among member models of an ensemble using its ensemble diversity score. (3) We introduce a two phase hierarchical pruning method to effectively identify and prune those deep ensembles with high HQ diversity scores, aiming to increase the lower and upper bounds on ensemble accuracy for the selected ensembles. By combining these three optimizations, deep ensembles selected based on our hierarchical diversity pruning approach significantly outperforms those selected by the corresponding Q-metrics. Comprehensive experimental evaluation over several benchmark datasets shows that our HQ-metrics can effectively select high diversity deep ensembles by pruning out those ensembles with insufficient diversity, and successfully increase the lower bound (worst case) accuracy of the selected deep ensembles, compared to those selected using the state-of-the-art Q-metrics.

## 1   INTRODUCTION

Deep ensembles with sufficient ensemble diversity hold potential of improving both accuracy and robustness of ensembles with their combined wisdom. The improvement can be measured by three criteria: (i) the average ensemble accuracy of the selected ensemble teams, (ii) the percentage of selected ensembles that exceed the highest accuracy of individual member models; (iii) the lower bound (worst case) and the upper bound (best case) accuracy of the selected ensembles. The higher these three measures, the higher quality of the ensemble teams. Ensemble learning can be broadly classified into two categories: (1) learning the ensemble of diverse models via diversity optimized joint-training, coined as the ensemble training approach, such as boosting (Schapire, 1999); and (2) learning to compose an ensemble of base models from a pool of existing pre-trained models through ensemble teaming based on ensemble diversity metrics (Partridge & Krzanowski, 1997; Liu et al., 2019; McHugh, 2012; Skalak, 1996), coined as the ensemble consensus approach. This paper is focused on improving the state-of-the-art results in the second category.

**Related Work and Problem Statement.** Ensemble diversity metrics are by design to capture the degree of negative correlation among the member models of an ensemble team (Brown et al., 2005; Liu et al., 2019; Kuncheva & Whitaker, 2003) such that the high diversity indicates high negative correlation among member models of an ensemble. Three orthogonal and yet complimentary

threads of efforts have been engaged in ensemble learning: (1) developing mechanisms to produce diverse base neural network models, (2) developing diversity metrics to select ensembles with high ensemble diversity from the candidate ensembles over the base model pool, and (3) developing consensus voting methods. The most popular consensus voting methods include the simple averaging, the weighted averaging, the majority voting, the plurality voting (Ju et al., 2017), and the learn to rank (Burges et al., 2005). For the base model selection, early efforts have been devoted to training diverse weak models to form a strong ensemble on a learning task, such as bagging (Breiman, 1996), boosting (Schapire, 1999), or different ways of selecting features, e.g., random forests (Tin Kam Ho, 1995). Several recent studies also produce diverse base models by varying the training hyper-parameters, such as snapshot ensemble (Huang et al., 2017), which utilizes the cyclic learning rates (Smith, 2015; Wu et al., 2019) to converge the single DNN model at different epochs to obtain the snapshots as the ensemble member models. Alternative method is to construct the pool of base models by using pre-trained models with different neural network backbones (Wu et al., 2020; Liu et al., 2019; Wei et al., 2020; Chow et al., 2019a). The research efforts on diversity metrics have proposed both pairwise and non-pairwise ensemble diversity measures (Fort et al., 2019; Wu et al., 2020; Liu et al., 2019), among which the three representative **pairwise** metrics are Cohen's Kappa (CK) (McHugh, 2012), Q Statistics (QS) (Yule, 1900), Binary Disagreement (BD) (Skalak, 1996), and the three representative **non-pairwise** diversity metrics are Fleiss' Kappa (FK) (Fleiss et al., 2013), Kohavi-Wolpert Variance (KW) (Kohavi & Wolpert, 1996; Kuncheva & Whitaker, 2003) and Generalized Diversity (GD) (Partridge & Krzanowski, 1997). These diversity metrics are widely used in several recent studies (Fort et al., 2019; Liu et al., 2019; Wu et al., 2020). Some early study has shown that these diversity metrics are correlated with respect to ensemble accuracy and diversity in the context of traditional machine learning models (Kuncheva & Whitaker, 2003). However, few studies to date have provided in-depth comparative critique on the effectiveness of these diversity metrics in pruning those low quality deep ensembles from the candidate ensembles due to their high negative correlation.

**Scope and Contributions.** In this paper, we focus on the problem of defining ensemble diversity metrics that can select diverse ensemble teams with high ensemble accuracy. We first investigate the six representative ensemble diversity metrics, coined as Q metrics. We identify and analyze their inherent limitations in capturing the negative correlation among the member models of an ensemble, and why pruning out those deep ensembles with low Q-diversity may not always guarantee to improve the ensemble accuracy. To address the inherent problems of Q metrics, we extend the existing six Q metrics with three optimizations: (1) We introduce the concept of the focal model and argue that one way to better capture the negative correlations among member models of an ensemble is to compute diversity scores for ensembles of fixed size based on the focal model. (2) We introduce the six HQ diversity metrics to optimize the six Q-diversity metrics respectively. (3) We develop a HQ-based hierarchical pruning method, consisting of two stage pruning: the $\alpha$ filter and the K-Means filter. By combining these optimizations, the deep ensembles selected by our HQ-metrics can significantly outperform those deep ensembles selected by the corresponding Q metrics, showing that the HQ metrics based hierarchical pruning approach is efficient in identification and removal of low diversity deep ensembles. Comprehensive experiments are conducted on three benchmark datasets: CIFAR-10 (Krizhevsky & Hinton, 2009), ImageNet (Russakovsky et al., 2015), and Cora (Lu & Getoor, 2003). The results show that our hierarchical diversity pruning approach outperforms their corresponding Q-metrics in terms of the lower bound and the upper bound of ensemble accuracy over the deep ensembles selected, exhibiting the effectiveness of our HQ approach in pruning low diversity deep ensembles.

## 2 HIERARCHICAL PRUNING WITH DIVERSITY METRICS

Existing studies on consensus based ensemble learning (Huang et al., 2017; Krizhevsky et al., 2012; Zoph & Le, 2016) generate the base model pool through two channels: (i) *deep neural network training* using different network structures or different configurations of hyperparameters (Breiman, 1996; Schapire, 1999; Zoph & Le, 2016; Hinton et al., 2015; Wu et al., 2018; 2019) and (ii) *selecting the top performing pre-trained models* from open-source projects (e.g., GitHub) and public model zoos (Jia et al., 2014; ONNX Developers, 2020; GTModelZoo Developers, 2020). Hence, an important technical challenge for deep ensemble learning is to define diversity metrics for producing high quality ensemble teaming strategies, aiming to boost the ensemble accuracy. Given that the number of possible ensemble teams increases exponentially with a small pool of base models, de-

veloping proper ensemble diversity metrics is critical for effective pruning of deep ensembles with insufficient diversity. Consider a pool of $M$ base models for a learning task on a given dataset $\mathcal{D}$, denoted by $BMSet(\mathcal{D})= \{F_1, ..., F_M\}$. Let $EnsSet$ denote the set of all possible ensemble teams that are composed from $BMSet(\mathcal{D})$, with the ensemble team size $S$ varying from 2 to $M$. We have a total of $\sum_{S=2}^{M} \binom{M}{S}$ ensembles, i.e., $|EnsSet| = \binom{M}{2} + \binom{M}{3} + ... + \binom{M}{M} = 2^M - (1 + M)$. The cardinality of the set of possible ensembles $EnsSet$ grows exponentially with $M$, the number of base models. For example, $M = 3$, we have $|EnsSet| = 4$. When $M$ becomes larger, such as $M = 5, 10, 20, |EnsSet| = 26, 1013, 1048555$. Hence, as $M$ increases, it is non-trivial to construct a set of high-accuracy ensemble teams ($GEnsSet$), from the candidate set ($EnsSet$) of all possible ensembles that are composed from $BMSet(\mathcal{D})$.

Consider a pool of $M = 10$ base models for ImageNet, in which the highest performing base model is 78.25%, the lowest performing base model is 56.63%, and the average accuracy of these 10 base models is 71.60% (see Table 5 in Appendix Section F). For a pool of 10 base models, there will be a total of 1013 ($2^{10} - (10 + 1)$) different ensembles with team size ranging from 2 to 10. The performance of these ensembles vary sharply, from 61.39% (lower bound) to 80.77% (upper bound). Randomly selecting an ensemble team from these 1013 teams in $EnsSet$(ImageNet) may lead to a non-trivial probability of selecting a team with the ensemble accuracy lower than the average member model accuracy of 71.60% over the 10 base models. Clearly, an efficient ensemble diversity metric should be able to prune out those ensemble teams with insufficient ensemble diversity and thus low ensemble accuracy, increasing (i) the **average ensemble accuracy** of the selected ensemble teams, (ii) the **percentage of selected ensembles** that exceed the highest accuracy of individual member models (i.e., 78.25% for the 10 base DNN models on ImageNet); and (iii) the **lower bound** (worst case) and the **upper bound** (best case) accuracy of the selected ensembles. A number of ensemble diversity metrics have been proposed to address this challenging problem. In this section, we first provide a comparative study of the six state-of-the-art Q-diversity metrics and analyze their inherent limitations in identifying and pruning out low diversity ensembles. Then we introduce our proposed HQ-diversity metrics and analyze the effectiveness of our HQ based hierarchical diversity approach in pruning low quality ensembles.

## 2.1 Q-DIVERSITY METRICS AND THEIR LIMITATIONS

We outline the key notations for the six Q-diversity metrics in Table 1: three pairwise diversity metrics: Cohen's Kappa (CK) (McHugh, 2012), Q Statistics (QS) (Yule, 1900) and Binary Disagreement (BD) (Skalak, 1996), and three non-pairwise diversity metrics: Fleiss' Kappa (Fleiss et al., 2013) (FK), Kohavi-Wolpert Variance (KW) (Kohavi & Wolpert, 1996; Kuncheva & Whitaker, 2003) and Generalized Diversity (GD) (Partridge & Krzanowski, 1997). The arrow column $\uparrow | \downarrow$ specifies the relationship between the Q-value and the ensemble diversity. The $\uparrow$ represents positive relationship of the Q-value and the ensemble diversity, that is a high Q-value refers to high ensemble diversity. The $\downarrow$ indicates the negative relationship, that is the low Q-value corresponds to high ensemble diversity. To facilitate the comparison of the six Q-diversity metrics such that the low Q-value refers to high ensemble diversity for all six Q-metrics, we apply ($1 - Q$-value) when calculating Q-diversity score with $BD$, $KW$ and $GD$. We refer readers to Appendix (Section C) for the formal definitions of the six Q-diversity metrics.

Table 1: The six Q-diversity metrics

| Category | Name | Notation | $\uparrow | \downarrow$ | Mean Threshold | | |
| --- | --- | --- | --- | --- | --- | --- |
| | | | | CIFAR-10 | ImageNet | Cora |
| Pairwise | Cohen's Kappa | $CK$ | $\downarrow$ | 0.562 | 0.500 | 0.468 |
| | Q Statistics | $QS$ | $\downarrow$ | 0.515 | 0.697 | 0.373 |
| | Binary Disagreement | $BD$ | $\uparrow$ | 0.661 | 0686 | 0.617 |
| Non-pawise | Fleiss' Kappa | $FK$ | $\downarrow$ | 0.561 | 0.499 | 0.461 |
| | Kohavi-Wolpert variance | $KW$ | $\uparrow$ | 0.868 | 0.878 | 0.851 |
| | Generalized Diversity | $GD$ | $\uparrow$ | 0.476 | 0.665 | 0.592 |

Given a Q-diversity metric, we calculate the diversity score for each ensemble team in the ensemble set ($EnsSet$) using a set of negative samples ($NegSampSet$) on which one or more models in the ensemble make prediction errors. The low Q-score indicates sufficient diversity among member models of an ensemble. Upon the completion of Q-diversity score computation for all ensembles in $EnsSet$, the diversity threshold based pruning is employed to remove those ensembles with insufficient diversity among ensemble member models. Either a pre-defined Q-diversity threshold or a

mean threshold by taking the average value of all Q-diversity scores calculated for all candidate deep ensembles in $EnsSet$. The mean threshold tends to work better in general than a manually defined threshold. Once a mean threshold is obtained, those ensembles in $EnsSet$ with their Q diversity scores below the threshold will be selected and placed into the diverse ensemble set $GEnsSet$, and the remaining ensembles are those with their Q scores higher than the threshold and thus will be pruned out. The pseudo code of the algorithm is included in Appendix (Algorithm 1). The last three columns of Table 1 show the mean threshold for all six Q-diversity metrics calculated on the set of 1013 candidate deep ensembles for the three benchmark datasets used in this study. We make two observations. First, different Q-diversity metrics capture the ensemble diversity from different perspectives with different diversity measurement principles, resulting in different Q-scores. Second, each Q-metric, say CK, is used to compare ensembles based on their Q-CK scores. Hence, even though the Q-KW metric has relatively high KW-specific Q scores for all ensemble teams, it can select the diverse ensembles based on the mean KW-threshold, in a similar manner as any of the other five Q metrics.

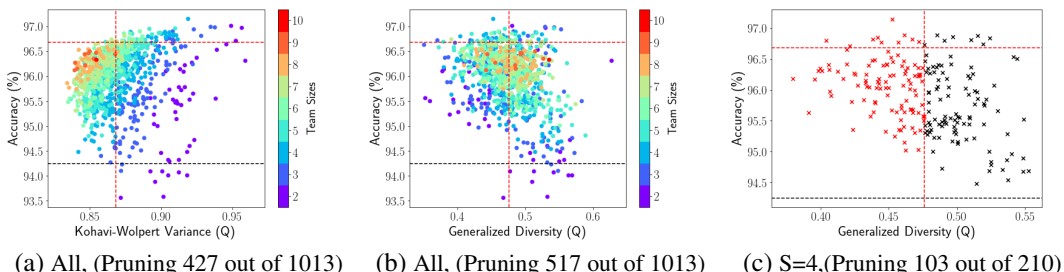

(a) All, (Pruning 427 out of 1013)  (b) All, (Pruning 517 out of 1013)  (c) S=4,(Pruning 103 out of 210)

Figure 1: Pruning with Q diversity with mean threshold (CIFAR-10))

**Limitations of Q Metrics.** Figure 1a and 1b show Q-KW and Q-GD metrics and their relationship with ensemble accuracy for all 1013 deep ensembles on CIFAR-10 respectively. Each dot represents a deep ensemble team with team sizes color-coded by the color diagram on the right. The vertical red dashed line represents the Q-KW and Q-GD mean thresholds of 0.868 and 0.476 respectively. The horizontal red and black dashed lines represent the maximum single model accuracy 96.68% and the average accuracy 94.25% of the 10 base models respectively. We use these two accuracy bounds as important references to quantify the quality of the deep ensembles selected using a Q metric with its mean threshold. Those deep ensembles on the left of the red vertical dash line are selected and added into $GEnsSet$ given that their Q-scores are below the mean threshold (e.g., Q-KW or Q-GD). The ensembles on the right of this red vertical dash line are pruned out because their Q diversity scores exceed the mean threshold. Compare Figure 1a and 1b, it is visually clear that both Q metrics can select a sufficient number of good quality ensemble teams while at the same time, both Q metrics with mean threshold pruning will miss a large portion of teams with high ensemble accuracy, indicating the inherent limitations of both Q metrics and the mean threshold pruning with respect to capturing the concrete ensemble diversity in terms of low negative correlation among member models of an ensemble.

To better understand the inherent problems with the Q-diversity metrics, we performed another set of experiments by measuring the Q-GD metric over ensemble teams of fixed size $S$ on CIFAR-10. Figure 1c shows a visualization of the results using the Q-GD scores computed over ensembles of size $S = 4$ with mean threshold indicated by the vertical red dashed line, showing a visually sharper trend in terms of the relationship between ensemble diversity and ensemble accuracy when comparing the selected ensemble teams (red dots) with those ensembles (black dots) on the right of the red vertical threshold line. However, relying on separating the diversity computation and comparison over ensemble teams of the same size alone may not be sufficient, because Figure 1c shows that (i) some selected ensemble teams have low accuracy, affecting all three ensemble quality measures (recall Section 2, page 3), and (ii) a fair number of ensemble teams with high ensemble accuracy (black dots on the top right side) are still missed. Similar observations are also found for other five Q-diversity metrics. We conclude our analysis with three arguments: (1) The Q-diversity metrics may not accurately capture the degree of negative correlation among the member models of an ensemble even when its ensemble Q-diversity score is below the mean threshold. (2) Comparing ensembles of different team size $S$ using their Q scores may not be a fair measure of their true ensemble diversity in terms of the degree of negative correlation among member models

of an ensemble. However, relying on ensembles of the same team size $S$ alone is still insufficient. (3) Mean threshold is not a good Q-diversity pruning method in terms of capturing the intrinsic relationship between ensemble diversity and ensemble accuracy. This motivates us to propose the HQ diversity metrics with two phase pruning using learning algorithms.

## 2.2 HQ-DIVERSITY METRICS AND THEIR TWO-PHASE PRUNING

The design of the six HQ metrics is to enhance the six existing popular Q-metrics with three optimizations. First, we argue that comparing ensembles of the same team size in terms of their diversity scores can better capture the intrinsic relationship between ensemble diversity and ensemble accuracy. Second, to further improve the comparison of ensembles of the same size $S$ in terms of their ensemble diversity in the context of negative correlation, we introduce the concept of focal model to obtain the set of negative samples for computing the diversity scores of ensembles by taking each member model in turn as the focal model. This is motivated by adversarial robustness with ensemble defense (Chow et al., 2019b; Wei & Liu, 2020), which composes robust ensemble teams for each attack target model. The concept of focal model allows us to capture the ensemble diversity of a team by utilizing the focal model and its negative samples, and then obtain a unified HQ score by taking an average of the $S$ focal model based diversity measurements for each ensemble team of size $S$. These two optimizations enable HQ scores to more accurately capture the ensemble diversity and its relationship with the ensemble accuracy. Finally, we employ the third optimization, which utilizes the two-phase HQ score based filtering process using the $\alpha$ filter and then the $K$-means filter to select the set of high quality ensembles.

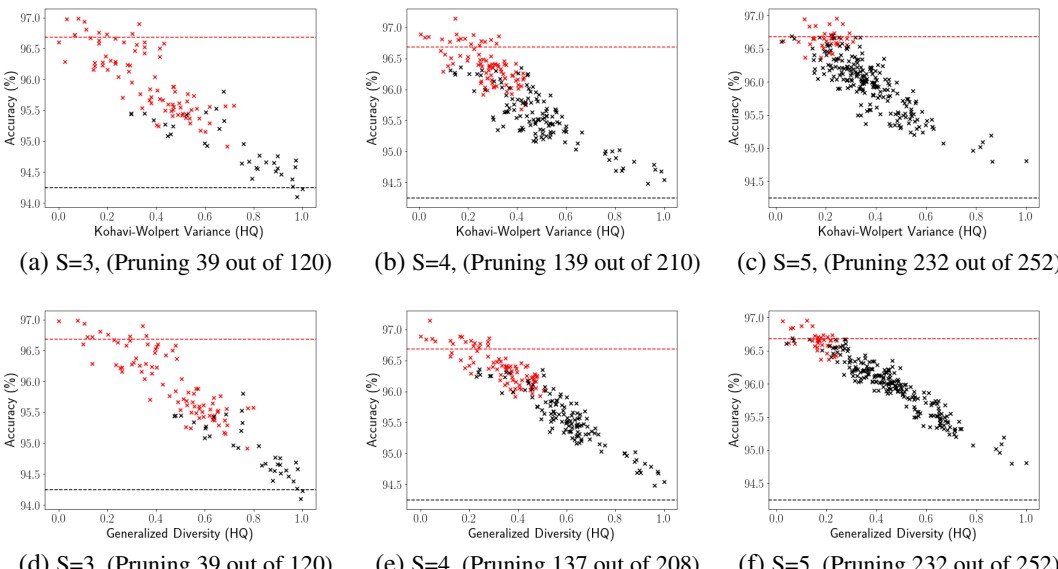

(a) S=3, (Pruning 39 out of 120)  (b) S=4, (Pruning 139 out of 210)  (c) S=5, (Pruning 232 out of 252)

(d) S=3, (Pruning 39 out of 120)  (e) S=4, (Pruning 137 out of 208)  (f) S=5, (Pruning 232 out of 252)

Figure 2: Ensemble teams of size $S = 3, 4, 5$ on CIFAR-10: top three figures for **HQ-KW** ($\alpha$) and bottom three figures for **HQ-GD** ($\alpha$)

**HQ ($\alpha$): HQ metrics with $\alpha$ filter.** We observe that if an ensemble team of size $S$ has large HQ score (say $[F_5, F_6]$), indicating insufficient ensemble diversity, then all the ensemble teams that have larger size than $S$ and contain all the member models of this ensemble team (e.g., $[F_5, F_6, F_7]$, $[F_0, F_5, F_6]$, $[F_0, F_5, F_6, F_7]$, $[F_5, F_6, F_7, F_8]$) tend to have insufficient ensemble diversity (i.e., larger HQ score) as well. This motivates us to design a hierarchical pruning algorithm, coined as $\alpha$ **filter**. Concretely, we start with the set of ensembles of smallest team size, say $S = 2$, $|EnsSet| = \binom{M}{2} = M(M - 1)$ candidate ensembles. For $M = 10$ we will have 90 teams of size 2. Given a HQ metric, we first sort the ensembles of small size $S$, say $S = 2$, by their HQ scores in decreasing order, and then choose the top $\beta$ (percentage) of ensembles of size $S$ with large HQ value as our pruning targets at team size $S$. We recommend a conservative approach by using a small $\beta$ (e.g., $\beta = 5\%, 10\%$). We first preemptively prune out the $\beta(\%)$ of the ensembles with large HQ scores and then prune all those ensembles that are super-sets of these $\beta(\%)$ of ensembles. Imagining a hierarchical structure with all teams of size 2 on the top, and each layer we add one

additional model to the teams such that all teams of size $S + 1$ are placed in the next tier. The bottom tier will be one ensemble team of size $M$. For each of the $\beta(\%)$ of ensembles of size 2 that are pruned out, this $\alpha$ filter algorithm will cut off the whole branch of ensemble teams that are supersets of this removed ensemble team. Due to space constraint, we include the Algorithm 2 to compute HQ metrics and the $\alpha$ filter algorithm in Appendix: section D and section E respectively.

Figure 2 shows the visualization of applying $\alpha$ filter on two HQ metrics: HQ-KW and HQ-GD. The black dots denote the ensemble teams pruned out by using the $\alpha$ filter and the red dots are the ensembles selected after HQ metric with $\alpha$ pruning. We highlight two interesting observations. *First,* the $\alpha$ filter can effectively prune those ensembles with large HQ values (representing insufficient ensemble diversity). Compared Q-GD in Figure 1c with HQ-GD ($\alpha$) in Figure 2e (both with $S = 4$), HQ-GD ($\alpha$) can significantly improve the quality of selected ensemble teams while effectively pruning out most of the low accuracy ensembles. *Second,* both HQ-GD ($\alpha$) and HQ-KW ($\alpha$) diversity metrics display high correlation of measured ensemble diversity with the ensemble accuracy: low HQ scores correspond to high ensemble accuracy. Similar observations are found consistently for all HQ diversity metrics.

**HQ ($\alpha + K$) metrics: HQ metrics with $\alpha$ filter followed by $K$-means filter.** In our two-phase HQ diversity pruning approach, we introduce $K$-means filter to correct as much as possible the remaining errors in high quality ensemble team selection. Recall Figure 2a and 2d for ensemble teams of size $S = 3$, it is visually clear that the $\alpha$ filter is less effective in pruning out some ensemble teams of low accuracy, compared to teams of larger sizes, $S = 4, 5$ in Figure 2(b)(c)(e)(f). We introduce the second phase filtering by using a customized $K$-means clustering algorithm with $K = 2$ and two strategically chosen initial centroids: top left and bottom right (marked in the red and black unfilled circles respectively), aiming to learn two clusters of ensembles: (1) the cluster of ensembles with low HQ score and high ensemble accuracy, and (2) the cluster of ensembles with low accuracy and relatively larger HQ score. The clustering results are indicated by the two solid circles: the pink one for cluster (1) and the light grey one for cluster (2). The two phase filtering powered HQ ($\alpha + K$) metrics can effectively remove those ensembles with low accuracy and insufficient diversity (i.e., higher HQ values), further improving the three ensemble accuracy measures (recall Section 2, page 3) compared to the HQ ($\alpha$) metrics, increasing the lower bound accuracy and improving the worst-case ensemble selection quality. Figure 3 provides a visualization for ensemble teams of size $S = 3$ using three HQ ($\alpha + K$) metrics: HQ-CK, HQ-KW and HQ-GD. The red dots and black dots show the two clusters produced by K-means, and the red vertical dashed line indicates the filtering threshold produced by the K-Means filter, which chooses the smallest HQ value from the cluster of low accuracy ensembles as the HQ-specific pruning threshold. By using HQ with two phase $\alpha + K$ filters, we can further fine tune the quality of ensemble selection by removing those ensembles with relatively low ensemble accuracy, effectively boosting the lower bound of ensemble accuracy for all the ensemble teams selected by HQ ($\alpha + K$) metrics, compared to either HQ ($\alpha$) or Q metrics.

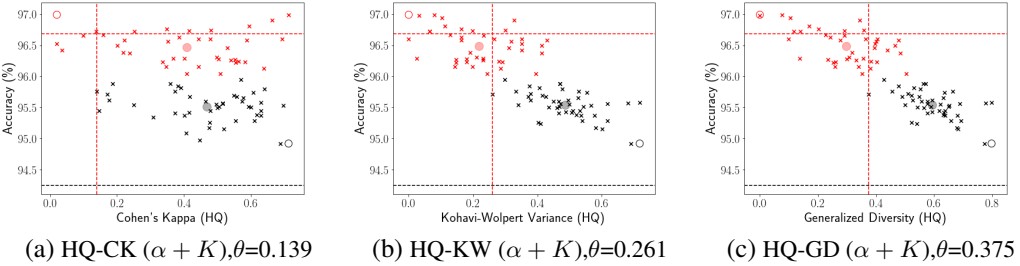

(a) HQ-CK ($\alpha + K$),$\theta$=0.139     (b) HQ-KW ($\alpha + K$),$\theta$=0.261     (c) HQ-GD ($\alpha + K$),$\theta$=0.375

Figure 3: Three HQ metrics with two phase ($\alpha + K$) filters for the team size $S = 3$ (CIFAR-10)

## 3 EXPERIMENTAL EVALUATION

Extensive experiments on three benchmark datasets (CIFAR-10, ImageNet, and Cora), with a total of 10 base models for each dataset, are conducted to evaluate our hierarchical diversity pruning methods. All the experiments were conducted on an Intel Xeon E5-1620 server with Nvidia GeForce GTX 1080Ti GPU on Ubuntu 16.04. Readers may refer to Appendix (section F) for further details on the base models used in this study and their accuracy results.

**CIFAR-10** Table 2 shows the experimental comparison of the ensemble teams selected by Q metrics with mean threshold, HQ ($\alpha$) metrics and HQ ($\alpha + K$) metrics for CIFAR-10. For the se-

lected ensembles, we show their ensemble accuracy range (%) in the 4th column. The 5th column #(%)(Acc>96.68% (max)) shows the number and percentage of the ensembles selected, which have ensemble accuracy higher than the highest (max) single model accuracy of 96.68% over the $M = 10$ CIFAR-10 base models. The last column shows the number of selected ensembles with ensemble accuracy over 96.70%, exceeding the best 96.68% single base model accuracy. We highlight three interesting observations. *First*, compare to Q metrics, our HQ ($\alpha$) metrics significantly reduce the number of candidate ensembles in #EnsSet (from 1013 to 230~281) and improve the quality of selected ensembles. For example, with the $\alpha$ filter, HQ-BD, HQ-KW and HQ-GD can improve the ensemble accuracy lower bound from 93.56% to 93.88%, while HQ-CK, HQ-QS, HQ-BD, HQ-FK and HQ-KW all improve the accuracy upper bound from 96.72% or 96.74% to 97.01% or 97.15%. *Second*, the two phase filtering HQ ($\alpha + K$) metrics further improved the quality of selected ensembles compared to both Q-metrics and HQ ($\alpha$) metrics, e.g., increasing the lower bound of ensemble accuracy from 93.56%~94.27% to 94.46%~95.45%. Furthermore, 42.22% (38 out of 90) of the ensembles selected by HQ-GD ($\alpha + K$) have the ensemble accuracy above 96.70%, showing that with random picking of an ensemble from the selected set ($GEnsSet$), HQ-GD has higher than 42% probability to choose an ensemble team with accuracy better than the max accuracy of the 10 single base models for CIFAR-10, compared to 17.93% by HQ-GD ($\alpha$) and 7.26% by Q-GD. This further demonstrates the effectiveness of our HQ ($\alpha + K$) metrics.

Table 2: Comparing Q, HQ ($\alpha$), HQ ($\alpha + K$) metrics on CIFAR-10

| Methods | #EnsSet | #GEnsSet | Ensemble Acc Range (%) | #(%)(Acc > 96.68% (max)) | #(Acc >= 96.70%) |
|---|---|---|---|---|---|
| Baseline | 1013 | 1013 | 93.56~97.15 | 66 (6.52%) | 56 |
| Q-CK | 1013 | 544 | 93.56~96.72 | 1 (0.18%) | 1 |
| Q-QS | 1013 | 516 | 93.56~96.74 | 4 (0.78%) | 3 |
| Q-BD | 1013 | 550 | 93.56~96.72 | 2 (0.36%) | 1 |
| Q-FK | 1013 | 541 | 93.56~96.72 | 1 (0.18%) | 1 |
| Q-KW | 1013 | 586 | 94.27~96.74 | 5 (0.85%) | 1 |
| Q-GD | 1013 | 496 | 93.56~97.15 | 36 (7.26%) | 32 |
| HQ-CK ($\alpha$) | 230 | 209 | 93.56~97.01 | 18 (8.61%) | 18 |
| HQ-QS ($\alpha$) | 235 | 212 | 94.01~97.01 | 9 (4.25%) | 9 |
| HQ-BD ($\alpha$) | 279 | 249 | 93.88~**97.15** | 43 (17.27%) | 43 |
| HQ-FK ($\alpha$) | 261 | 235 | 93.56~97.01 | 25 (10.64%) | 25 |
| HQ-KW ($\alpha$) | 279 | 249 | 93.88~**97.15** | 43 (17.27%) | 43 |
| HQ-GD ($\alpha$) | 281 | 251 | 93.88~**97.15** | 45 (17.93%) | 44 |
| HQ-CK ($\alpha + K$) | 209 | 53 | **95.04**~97.01 | 6 (11.32%) | 6 |
| HQ-QS ($\alpha + K$) | 212 | 31 | **95.45**~96.73 | 2 (6.45%) | 2 |
| HQ-BD ($\alpha + K$) | 249 | 76 | **95.23**~97.15 | 26 (34.21%) | 26 |
| HQ-FK ($\alpha + K$) | 235 | 50 | **94.46**~97.01 | 5 (10.00%) | 5 |
| HQ-KW ($\alpha + K$) | 249 | 74 | **95.23**~97.15 | 27 (36.49%) | 27 |
| HQ-GD ($\alpha + K$) | 251 | 90 | **94.72**~97.15 | **38 (42.22%)** | 38 |

**ImageNet** Table 3 shows the same set of experiments on ImageNet. We make three observations. (1) For ImageNet, many ensembles generated by HQ metrics can achieve higher ensemble accuracy, better than the max single base model accuracy of 78.25% by the member model $F_5$ (Table 5 in Appendix Section F), even without having $F_5$ as a member model of the ensemble teams. For example, with $\alpha + K$, HQ-BD and HQ-GD both have 19 ensemble teams that offer ensemble accuracy higher than the max single model accuracy of 78.25% by the member model $F_5$, and yet do not have $F_5$ as the member model of their ensemble teams. (2) Similar to CIFAR-10, many ensembles with low accuracy and insufficient HQ diversity are effectively pruned out by using our HQ ($\alpha$) metrics. Compared to Q-metrics, our HQ ($\alpha$) metrics effectively increase the accuracy lower bound of all selected ensembles from 61.39% to 68.99%, significant improvement over Q metrics. (3) The HQ ($\alpha + K$) metrics further boost the lower bound ensemble accuracy over the corresponding HQ ($\alpha$) metrics, with the lower bound (worst case) accuracy of 76.16%~78.35%, significantly higher than Q metrics (61.39%~70.79%). Three HQ ($\alpha + K$) metrics (HQ-CK, HQ-QS, HQ-FK) achieve 100% of the selected ensembles with over 78.25% accuracy (the max single base model accuracy on ImageNet), while HQ-BD has over 90.91%, HQ-KW and HQ-GD have over 87.10% of the selected ensembles with their ensemble accuracy over the best single base model accuracy (78.25%). Clearly, the average accuracy of the selected ensembles by HQ ($\alpha + K$) metrics is much higher than that by using Q-diversity metrics.

Table 3: Comparing Q, HQ ($\alpha$), and HQ ($\alpha + K$) metrics on ImageNet

| Methods | #EnsSet | #GEnsSet | Ensemble Acc Range (%) | #(%)(Acc > 78.25% (max)) | #(Acc >= 79.50%) |
|---|---|---|---|---|---|
| Baseline | 1013 | 1013 | 61.39~80.77 | 753 (74.33%) | 343 |
| Q-CK | 1013 | 555 | 61.39~80.50 | 338 (60.90%) | 92 |
| Q-QS | 1013 | 483 | 61.39~80.54 | 296 (61.28%) | 96 |
| Q-BD | 1013 | 554 | 61.39~80.54 | 349 (63.00%) | 107 |
| Q-FK | 1013 | 553 | 61.39~80.50 | 336 (60.76%) | 91 |
| Q-KW | 1013 | 647 | 68.72~80.56 | 473 (73.11%) | 188 |
| Q-GD | 1013 | 530 | 70.79~80.60 | 394 (74.34%) | 170 |
| HQ-CK ($\alpha$) | 221 | 201 | 68.99~80.42 | 94 (46.77%) | 16 |
| HQ-QS ($\alpha$) | 245 | 221 | 70.80~80.70 | 157 (71.04%) | 70 |
| HQ-BD ($\alpha$) | 283 | 253 | 69.21~**80.77** | 198 (78.26%) | 110 |
| HQ-FK ($\alpha$) | 221 | 201 | 68.99~80.42 | 94 (46.77%) | 16 |
| HQ-KW ($\alpha$) | 283 | 253 | 69.21~**80.77** | 198 (78.26%) | 110 |
| HQ-GD ($\alpha$) | 287 | 257 | 69.21~**80.77** | 203 (78.99%) | 115 |
| HQ-CK ($\alpha + K$) | 201 | 4 | **78.28**~79.64 | **4 (100.00%)** | 1 |
| HQ-QS ($\alpha + K$) | 221 | 16 | **78.35**~80.54 | **16 (100.00%)** | 9 |
| HQ-BD ($\alpha + K$) | 253 | 55 | **77.18**~80.77 | 50 (90.91%) | 36 |
| HQ-FK ($\alpha + K$) | 201 | 4 | **78.28**~79.64 | **4 (100.00%)** | 1 |
| HQ-KW ($\alpha + K$) | 253 | 65 | **76.16**~80.77 | 57 (87.69%) | 44 |
| HQ-GD ($\alpha + K$) | 257 | 62 | **76.16**~80.77 | 54 (87.10%) | 40 |

**Ensemble Accuracy Distribution.** We further investigate the ensemble accuracy distribution for the ensemble teams selected by Q, HQ ($\alpha$) and HQ ($\alpha + K$) metrics. Figure 4 shows the visualization of the results. For CIFAR-10, we compare the ensemble teams selected by Q-GD (yellow triangles), HQ-GD ($\alpha$) (blue dots), and HQ-GD ($\alpha + K$) (red circles). It is visually clear that HQ-GD ($\alpha + K$) diversity metric can effectively prune out more low accuracy ensembles with insufficient HQ scores compared to Q-GD and HQ-GD ($\alpha$), although it still suffers from a few low accuracy ensembles, which dragged the improvement on the ensemble accuracy lower bound of 94.72% on CIFAR-10. For ImageNet, Figure 4b and 4c show that both HQ-BD ($\alpha + K$) and HQ-GD ($\alpha + K$) have the best performance with most of the selected ensembles on the top left (red circles), indicating high ensemble accuracy and high lower bound on the ensemble accuracy of all selected teams.

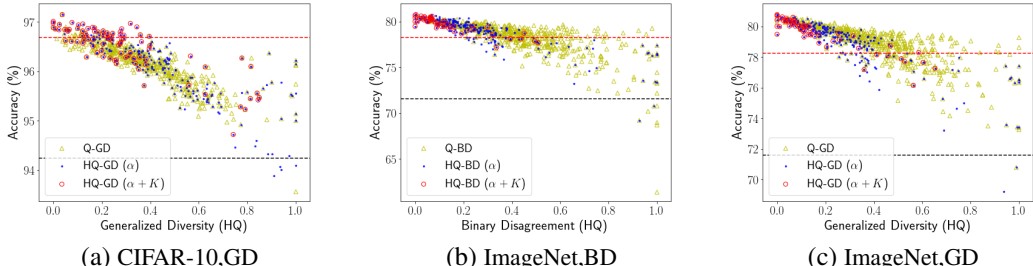

(a) CIFAR-10,GD      (b) ImageNet,BD      (c) ImageNet,GD

Figure 4: Ensemble Accuracy Distribution on CIFAR-10 and ImageNet

## 4 CONCLUSION

We have presented a two-phase hierarchical ensemble diversity pruning approach for high quality ensemble selection. This paper makes three original contributions. First, we identify and analyze the inherent limitations of existing six ensemble diversity metrics, coined as Q-metrics. Second, we address the limitations of Q-metrics by introducing the six HQ diversity metrics respectively. Third, we develop a two phase HQ-based hierarchical pruning method with $\alpha$ filter followed by $K$-means filter. By combining these optimizations, the deep ensembles selected by our HQ ($\alpha + K$) metrics can significantly outperform the deep ensembles selected by the corresponding Q metrics, showing that the HQ metrics based hierarchical pruning approach is efficient in identification and removal of low quality deep ensembles. Comprehensive experiments conducted on benchmark datasets of CIFAR-10 and ImageNet show that our hierarchical diversity pruning approach outperforms the corresponding Q-metrics in terms of the **lower bound** (worst case) and the **upper bound** (best case) of ensemble accuracy over the deep ensembles selected, in addition to the **average ensemble accuracy** of the selected ensemble teams, and the **percentage of selected ensembles** that exceed the highest accuracy of the member models in the base model pool.

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

## A  DIVERSITY BY UNCORRELATED ERROR

Deep neural network ensembles use multiple (say $M > 1$) deep neural networks to form a committee (team) to collaborate and combine the predictions of individual member models to make the final prediction. A consensus method will be used to combine the individual predictions, such as majority voting, plurality voting, or model averaging (the average of prediction vectors).

A deep neural network classifier is typically trained to minimize a cross-entropy loss and output a probability vector to approximate a posteriori probability densities for the corresponding class. For a given input $x$, the $i$th element in the output probability vector of model $F_k$ can be modeled as: $f_i^k(x) = p(c_i|x) + \epsilon_i^k(x)$, where $p(c_i|x)$ is the posteriori probability distribution of the $i$th class ($c_i$) for the input $x$, and $\epsilon_i^k(x)$ is the error associated with this output. For making the Bayes optimum decision, $x$ will be predicted as class $c_i$ if $p(c_i|x) > p(c_j|x), \forall j \neq i$. Therefore, the Bayes optimum boundary locates at all points $x^*$ such that $p(c_i|x^*) = p(c_j|x^*)$ where $p(c_j|x^*) = max_{l \neq i} p(c_l|x)$. Given the neural network model will output $f_i^k(x)$ instead of $p(c_i|x)$, the decision boundary of the model, $\bar{x}$, may vary from the optimum boundary by an offset $o = \bar{x} - x^*$. (TUMER & GHOSH, 1996) shows that the added error beyond Bayes error is $E_{add} = \frac{d\sigma_o^2}{2}$ where $d$ is the difference between the derivatives of the two posteriors and $\sigma_o^2$ is the variance of the boundary offset $o$, $\sigma_o^2 = 2\sigma_{\epsilon_i^k}^2/d^2$. Combining the predictions of $S$ models with model averaging ($avg$), the $i$th element in the combined probability vector gives an approximation to $p(c_i|x)$ as $f_i^{avg}(x) = \frac{1}{S}\sum_{k=1}^{S} f_i^k(x) = p(c_i|x) + \bar{\epsilon}_i(x)$, where $\bar{\epsilon}_i(x) = \frac{1}{S}\epsilon_i^k(x)$. We can calculate the variance of $\bar{\epsilon}_i$ with

$$\sigma_{\bar{\epsilon}_i}^2 = \frac{1}{S^2}\sum_{k=1}^{S}\sum_{l=1}^{S} cov(\epsilon_i^k(x), \epsilon_i^l(x)) = \frac{1}{S^2}\sum_{k=1}^{S}\sigma_{\epsilon_i^k}^2 + \frac{1}{S^2}\sum_{k=1}^{S}\sum_{l\neq k} cov(\epsilon_i^k(x), \epsilon_i^l(x))$$

where $cov()$ represents the covariance. With $cov(a,b) = corr(a,b)\sigma_a\sigma_b$, we can replace the covariance with correlation $corr()$ and derive

$$\sigma_{\bar{\epsilon}_i}^2 = \frac{1}{S^2}\sum_{k=1}^{S}\sigma_{\epsilon_i^k}^2 + \frac{1}{S^2}\sum_{k=1}^{S}\sum_{l\neq k} corr(\epsilon_i^k(x), \epsilon_i^l(x))\sigma_{\epsilon_i^k}\sigma_{\epsilon_i^l}$$

Let $\delta_i$ denote the average correlation factor among these models, we have

$$\delta_i = \frac{1}{S(S-1)}\sum_{k=1}^{S}\sum_{l\neq k} corr(\epsilon_i^k(x), \epsilon_i^l(x))$$

Assuming the common variance $\sigma_{\epsilon_i}^2 = \sigma_{\epsilon_i^k}^2$ holds for every model $F_k$, with $\delta_i$ we have

$$\sigma_{\bar{\epsilon}_i}^2 = \frac{1}{S}\sigma_{\epsilon_i}^2 + \frac{S-1}{S}\delta_i\sigma_{\epsilon_i}^2$$

With the variance of the ensemble decision boundary offset $\sigma_{o^{avg}}^2 = \frac{\sigma_{\epsilon_i}^2 + \sigma_{\epsilon_j}^2}{d^2}$ given in (TUMER & GHOSH, 1996), we have

$$\sigma_{o^{avg}}^2 = \frac{1}{d^2 S}(\sigma_{\epsilon_i}^2 + (S-1)\delta_i\sigma_{\epsilon_i}^2 + \sigma_{\epsilon_j}^2 + (S-1)\delta_j\sigma_{\epsilon_j}^2)$$

Assume that the error between classes are i.i.d., that is $\sigma_{\epsilon_i}^2 = \sigma_{\epsilon_j}^2$. With $\sigma_{\epsilon_i}^2 = \sigma_{\epsilon_i^k}^2$ (the previous assumption) and $\sigma_o^2 = \frac{2\sigma_{\epsilon_i^k}^2}{d^2}$ given in (TUMER & GHOSH, 1996), we have

$$\sigma_{o^{avg}}^2 = \frac{1}{d^2 S}(2\sigma_{\epsilon_i}^2 + (S-1)\sigma_{\epsilon_i}^2(\delta_i + \delta_j))$$

$$\sigma_{o^{avg}}^2 = \frac{2\sigma_{\epsilon_i}^2}{d^2 S}(1 + (S-1)\frac{(\delta_i + \delta_j)}{2}) = \frac{2\sigma_{\epsilon_i^k}^2}{d^2 S}(1 + (S-1)\frac{(\delta_i + \delta_j)}{2})$$

$$\sigma_{o^{avg}}^2 = \frac{\sigma_o^2}{S}(1 + (S-1)\frac{\delta_i + \delta_j}{2})$$

To extend the above formula to include all classes, given $\delta = \sum_{i=1}^{C} P_i \delta_i$, where $P_i$ is the prior probability of class $c_i$ and $C$ is the total number of classes. Assuming the prior probability $P_i$ of class $c_i$ is uniformly distributed, we have

$$\sigma_{o^{avg}}^2 = \frac{\sigma_o^2}{S}(1 + (S-1)\delta) = \sigma_o^2(\frac{1 + (S-1)\delta}{S})$$

So we can derive the added error for the ensemble prediction $E_{add}^{avg}$ as

$$E_{add}^{avg} = \frac{d\sigma_{o^{avg}}^2}{2} = \frac{d\sigma_o^2}{2}(\frac{1 + (S-1)\delta}{S}) = E_{add}(\frac{1 + (S-1)\delta}{S})$$

Therefore, the ideal scenario is when all members in an ensemble team of size $S$ are diverse. They can learn and predict with uncorrelated errors (negative correlation), i.e., $\delta = 0$. Then a simple model averaging method can significantly reduce the overall prediction error by $S$. Meanwhile, the worst scenario happens when error of individual model are highly correlated with $\delta = 1$, such as all $S$ models are perfect duplicates, the error of the ensemble is identical to the initial errors without any improvement. In general, the correlation $\delta$ lies between 0 and 1, and therefore, it is always beneficial to use ensemble to reduce the prediction errors.

## B  ENSEMBLE ROBUSTNESS

Let $g(x) = f_c(x) - f_j(x)$, where $c = argmax_{1 \leq i \leq C} f_i(x)$ is the predicted class label and $j \neq c$. Assume $g(x)$ is Lipschitz continuous with Lipschitz constant $L_q^j$, according to (Paulavičius & Žilinskas, 2006), we have

$$|g(x) - g(y)| \leq L_q^j ||x - y||_p$$

where $L_q^j = max_x ||\nabla g(x)||_q$, $\frac{1}{p} + \frac{1}{q} = 1$ and $1 \leq p, q \leq \infty$.

Let $x = x_0 + \delta$ and $y = x_0$, we have

$$|g(x_0 + \delta) - g(x_0)| \leq L_q^j ||\delta||_p$$

which can be rearranged as

$$g(x_0) - L_q^j ||\delta||_p \leq g(x_0 + \delta) \leq g(x_0) + L_q^j ||\delta||_p$$

When $g(x_0 + \delta) = 0$, the predicted class label will change. However, $g(x_0 + \delta)$ is lower bounded by $g(x_0) - L_q^j ||\delta||_p \leq g(x_0 + \delta)$. If $0 \leq g(x_0) - L_q^j ||\delta||_p$, we have $g(x_0 + \delta) \geq 0$ to ensure that the prediction result will not change with the small change $\delta$ on the input $x_0$. This leads to

$$g(x_0) - L_q^j ||\delta||_p \geq 0 \Rightarrow ||\delta||_p \leq \frac{g(x_0)}{L_q^j}$$

That is

$$||\delta||_p \leq \frac{f_c(x_0) - f_j(x_0)}{L_q^j}$$

To ensure the classification result will not change, that is $argmax_{1 \leq i \leq C} f_i(x_0 + \delta) = c$, we use the minimum of the bound on $\delta$ over $j \neq c$, that is

$$||\delta||_p \leq min_{j \neq c} \frac{f_c(x_0) - f_j(x_0)}{L_q^j}$$

which indicates that as long as $||\delta||_p$ is small enough to fulfill the above bound, the classifier decision will never be changed, which marks the robustness of this classifier. The robustness bound ($R$) can be denoted as

$$R = min_{j \neq c} \frac{f_c(x_0) - f_j(x_0)}{L_q^j} = min_{j \neq c} \frac{f_c(x_0) - f_j(x_0)}{max_x ||\nabla(f_c(x) - f_j(x))||_q}$$

For a model $F_k$, we have its upper bound

$$R^k = min_{j \neq c} \frac{f_c^k(x_0) - f_j^k(x_0)}{max_x ||\nabla(f_c^k(x) - f_j^k(x))||_q}$$

Let $g_j^k(x) = f_c^k(x) - f_j^k(x)$, we have

$$R^k = min_{j \neq c} \frac{g_j^k(x_0)}{max_x ||\nabla(g_j^k(x))||_q}$$

Given $S$ models, combining their predictions with model averaging ($avg$), we have the $i$th element in the combined probability vector as $f_i^{avg}(x) = \frac{1}{S} \sum_{k=1}^{S} f_i^k(x)$ corresponding to the robustness bound

$$R^{avg} = min_{j \neq c} \frac{f_c^{avg}(x_0) - f_j^{avg}(x_0)}{max_x ||\nabla(f_c^{avg}(x) - f_j^{avg}(x))||_q} = min_{j \neq c} \frac{g_c^{avg}(x_0)}{max_x ||\nabla(g_c^{avg}(x))||_q}$$

Assume the minimum of the robustness bound can be achieved with the prediction result $c$ and $j$ for each model including the ensemble $F^{avg}$, that is

$$R^k = \frac{g_j^k(x_0)}{max_x ||\nabla(g_j^k(x))||_q}$$

and

$$R^{avg} = \frac{g_j^{avg}(x_0)}{max_x ||\nabla(g_j^{avg}(x))||_q}$$

where $g_j^{avg}(x) = \frac{1}{S} \sum_{k=1}^{S} g_j^k(x)$. The following property always holds that $\exists 1 \leq k \leq S, R^k \leq R^{avg}$, indicating that the ensemble can improve the robustness bound.

We prove the property by contradiction. First, we assume $\forall 1 \leq k \leq S, R^k > R^{avg}$, that is

$$\frac{g_j^k(x_0)}{max_x ||\nabla(g_j^k(x))||_q} > \frac{g_j^{avg}(x_0)}{max_x ||\nabla(g_j^{avg}(x))||_q}$$

So we have

$$g_j^k(x_0)(max_x ||\nabla(g_j^{avg}(x))||_q) > g_j^{avg}(x_0)(max_x ||\nabla(g_j^k(x))||_q)$$

For each $k \in \{1, ..., S\}$, we have the above inequality. To add them all, we have

$$\sum_{k=1}^{S} g_j^k(x_0)(max_x||\nabla(g_j^{avg}(x))||_q) > \sum_{k=1}^{S} g_j^{avg}(x_0)(max_x||\nabla(g_j^k(x))||_q)$$

That is

$$(max_x||\nabla(g_j^{avg}(x))||_q) \sum_{k=1}^{S} g_j^k(x_0) > g_j^{avg}(x_0) \sum_{k=1}^{S} (max_x||\nabla(g_j^k(x))||_q)$$

Given $g_j^{avg}(x) = \frac{1}{S} \sum_{k=1}^{S} g_j^k(x)$, we have

$$(max_x||\nabla(\sum_{k=1}^{S} g_j^k(x))||_q)\frac{1}{S}\sum_{k=1}^{S} g_j^k(x_0) > \frac{1}{S}\sum_{k=1}^{S} g_j^k(x_0) \sum_{k=1}^{S}(max_x||\nabla(g_j^k(x))||_q)$$

Therefore, we have

$$(max_x||\nabla(\sum_{k=1}^{S} g_j^k(x))||_q) > \sum_{k=1}^{S}(max_x||\nabla(g_j^k(x))||_q)$$

According to the triangle inequality, we have

$$max_x||\nabla(\sum_{k=1}^{S} g_j^k(x))||_q \leq max_x(\sum_{k=1}^{S} ||\nabla(g_j^k(x))||_q) \leq \sum_{k=1}^{S}(max_x||\nabla(g_j^k(x))||_q)$$

which contradicts with our derived inequality. Therefore, the previous assumption does not hold. We show that $\exists 1 \leq k \leq S, R^k \leq R^{avg}$, demonstrating that the robustness of a member model can be further improved with ensemble.

Furthermore, for a model $F^k$, if its robustness bound $R^k$ was not obtained with $j$. We have $\exists i \neq j, i, j \neq c, R^k = \frac{g_i^k(x_0)}{max_x||\nabla(g_i^k(x))||_q} \leq \frac{g_j^k(x_0)}{max_x||\nabla(g_j^k(x))||_q}$. The above claim still holds as long as each model makes the same prediction $c$.

## C  ALGORITHMS FOR COMPUTING Q-DIVERSITY METRICS

We have covered six state-of-the-art diversity metrics (coined in this paper as Q-diversity metrics). In the literature, different studies will use one of these diversity metrics to select models and analyze the prediction results. However, there are few studies to provide guidelines for choosing them or to compare and evaluate these diversity metrics in terms of pruning out low diversity ensembles.

In general, diversity metrics can be classified into two major categories based on how the fault independence and uncorrelated errors are computed using a set of negative samples. They are pairwise metrics and non-pairwise metrics. We below describe six representative diversity metrics considered in our study: Cohen's Kappa, Q Statistics and Binary Disagreement for pairwise, and Fleiss' Kappa, Kohavi-Wolpert Variance and Generalized Diversity for non-pairwise.

Given a pool of $M$ base models, all trained on the same dataset, one approach to create negative samples is to get the negative samples from the validation set of each model and then randomly select a subset of negative samples from the union of all $M$ subsets of negative examples. Let $\mathbf{X} = \{\mathbf{x}_1, \mathbf{x}_2, ..., \mathbf{x}_N\}$ be the randomly selected $N$ labeled negative examples on the training dataset. Given a base model $F_i$ and a negative sample $\mathbf{X}$, the output of $F_i$ on $\mathbf{X}$ is a vector of binary values, denoted as $\boldsymbol{\omega_i} = [\omega_{i,1}, \omega_{i,2}, ..., \omega_{i,N}]^T$, and $\omega_{i,k} = 1$ if $F_i$ predicts $\mathbf{x}_k$ correctly, otherwise, $\omega_{i,k} = 0$.

**Pairwise Diversity Metrics** For pairwise diversity metrics, they are calculated based on a pair of classifiers. Table 4 shows the relationship between a pair of classifiers $F_i, F_j$. For a labeled sample $\mathbf{x}_k$, four different types of prediction results emerge, such as both $F_i$ and $F_j$ make correct or wrong predictions and either $F_i$ or $F_j$ makes correct predictions. Correspondingly, we can count the number of samples in the four different types, that is $N^{ab}$, which represents the number of elements $\mathbf{x}_k \in \mathbf{X}$, such that $\omega_{i,k} = a$ and $\omega_{j,k} = b$.

Table 4: The relationship between a pair of classifiers

|  | $F_j$ correct (1) | $F_j$ wrong (0) |
|---|---|---|
| $F_i$ correct (1) | $N^{11}$ | $N^{10}$ |
| $F_i$ wrong (0) | $N^{01}$ | $N^{00}$ |
| $N = N^{00} + N^{01} + N^{10} + N^{11}$ | | |

**i. Cohen's Kappa (CK):** The Cohen's Kappa measures the diversity between the two classifiers $F_i, F_j$ from the perspective of agreement (McHugh, 2012; Kuncheva & Whitaker, 2003). A lower Cohen's kappa value implies lower agreement and higher diversity. Formula 1 shows the definition of the Cohen's kappa ($\kappa_{ij}$) between the two classifiers $F_i, F_j$. The value for the Cohen's Kappa ranges from -1 to 1 with 0 representing the amount of agreement of random chance. (McHugh, 2012)

$$\kappa_{ij} = \frac{2(N^{11}N^{00} - N^{01}N^{10})}{(N^{11} + N^{10})(N^{01} + N^{00}) + (N^{11} + N^{01})(N^{10} + N^{00})} \quad (1)$$

**ii. Q Statistics (QS):** The Q statistics (Yule, 1900) is defined as $QS_{ij}$ in Formula 2 between two models $F_i, F_j$. $QS_{ij}$ varies between -1 and 1. When the models $F_i, F_j$ are statistically independent, the expected $QS_{ij}$ is 0. If the two models tend to recognize the same object similarly, $QS_{ij}$ will have positive value. While two diverse models, recognizing the same object differently, will render a small or negative $QS_{ij}$ value.

$$QS_{ij} = \frac{N^{11}N^{00} - N^{01}N^{10}}{N^{11}N^{00} + N^{01}N^{10}} \quad (2)$$

**iii. Binary Disagreement (BD):** The binary disagreement (Skalak, 1996; Kuncheva & Whitaker, 2003) is the ratio between (i) the number of samples on which one model is correct while the other is wrong to (ii) the total number of samples predicted by the two models $F_i, F_j$ as Formula 3 shows.

$$\theta_{ij} = \frac{N^{01} + N^{10}}{N^{11} + N^{10} + N^{01} + N^{00}} \quad (3)$$

For an ensemble team of $S$ models, as recommended by (Kuncheva & Whitaker, 2003), we calculate the averaged metric value over all pair of classifiers as Formula 4 shows, where Q represents a pairwise diversity metric.

$$Q = \frac{2}{S(S-1)} \sum_{i=1}^{S-1} \sum_{j=i+1}^{S} Q_{ij} \quad (4)$$

**Non-pairwise Diversity Metrics** Numerous non-pairwise diversity metrics are widely used for a team of over 2 models. To compare with pairwise diversity metrics, we focus on three representative non-pairwise diversity metrics.

In an ensemble team of $S$ classifiers, we use $l(\mathbf{x}_k)$ to denote the number of classifiers that correctly recognize $\mathbf{x}_k$, i.e., $l(\mathbf{x}_k) = \sum_{i=1}^{S} \omega_{ik}$.

**iv. Fleiss' Kappa (FK):** Similar to Cohen's Kappa, the Fleiss' Kappa (Fleiss et al., 2013) also measures the diversity from the perspective of agreement. But it is directly calculated from a team of more than 2 models as Formula 5 shows, where $\bar{p}$ is the average classification accuracy for the

ensemble team and $\kappa$ is not obtained by simply averaging the Cohen's kappa ($\kappa_{ij}$).

$$\bar{p} = \frac{1}{NS} \sum_{k=1}^{N} \sum_{i=1}^{S} \omega_{i,k}$$

$$\kappa = 1 - \frac{\frac{1}{S} \sum_{k=1}^{N} l(\mathbf{x}_k)(S - l(\mathbf{x}_k)}{N(S-1)\bar{p}(1-\bar{p})} \tag{5}$$

**v. Kohavi-Wolpert Variance (KW):** Kohavi-Wolpert Variance is derived by (Kuncheva & Whitaker, 2003) to measure the variability of the predicted class label for the sample $\mathbf{x}$ with the team of models $F_1, F_2, ..., F_S$ as Formula 6 shows. Higher value of KW variance indicates higher model diversity of the team.

$$KW = \frac{1}{NS^2} \sum_{k=1}^{N} l(\mathbf{x}_k)(S - l(\mathbf{x}_k)) \tag{6}$$

**vi. Generalized Diversity (GD):** The generalized diversity has been proposed by (Partridge & Krzanowski, 1997) as Formula 7 shows. $Y$ is a random variable, representing the proportion of classifiers (out of $S$) that fail to recognize a random sample $\mathbf{x}$. The probability of $Y = \frac{i}{S}$ is denoted as $p_i$, i.e., the probability of $i$ (out of $S$) classifiers recognizing a randomly chosen sample $\mathbf{x}$ incorrectly. $p(1)$ represented the expected probability of one randomly picked model failing while $p(2)$ denotes the expected probability of both two randomly picked models failing. $GD$ varies between 0 and 1. The maximum diversity (1) occurs when the failure of one model is accompanied by the correct recognition by the other model for two randomly picked models, that is $p(2) = 0$. When both two randomly picked models fail, we have $p(1) = p(2)$, corresponding to the minimum diversity, 0.

$$p(1) = \sum_{i=1}^{S} \frac{i}{S} p_i$$

$$p(2) = \sum_{i=1}^{S} \frac{i(i-1)}{S(S-1)} p_i \tag{7}$$

$$GD = 1 - \frac{p(2)}{p(1)}$$

Algorithm 1 shows the sketch of the process of using a threshold-based filter. The diversity threshold calculation function is denoted as $\Theta$, such as the mean function. First, we calculate the diversity measurements for all ensemble teams. Then based on the diversity threshold $\theta(Q)$ (Line 10), we can prune out the teams with low diversity ($q_i \geq \theta(Q)$) and place the remaining high diversity ensembles into $GEnsSet$ (Line 11~15). With a proper threshold, $\theta$, the threshold-based pruning can efficiently prune out low-diversity deep ensembles.

## D   THE ALGORITHM FOR COMPUTING HQ-DIVERSITY METRICS

Unlike the Q-diversity metrics, HQ-diversity metrics calculate the diversity among the ensembles of the same size with a focal model. Algorithm 2 shows the skeleton of calculating the HQ diversity metrics for all the candidate ensembles in $EnsSet$. For each team size $S$ (Line 6~29), we follow two general steps to calculate the HQ diversity scores for each ensemble. *First,* each model in the base model pool will serve as the focal model. For the specific focal model $F_{focal}$, let $EnsSet(F_{focal}, S)$ denote all candidate ensembles of size $S$, each containing the member model $F_{focal}$. We first compute the Q-diversity score for each ensemble in $EnsSet(F_{focal}, S)$ with the negative samples drawn from the focal model $F_{focal}$ and store them in $D(Q, S, F_{focal})$ (Line 10~10). Then, in order to make them comparable across different focal models, we scale $D(Q, S, F_{focal})$ into $[0, 1]$ and store them into $\overline{D}(Q, S, F_{focal}, T_i)$ for each ensemble $T_i$ (Line 15~18). *Second,* for each candidate ensemble ($T_i$) of size $S$, we perform a weighted average of the scaled diversity scores $\overline{D}(Q, S, F_{focal} = T_i[j], T_i)$ associated with each of its member model $T_i[j]$ to obtain the unified

---

**Algorithm 1** Threshold-based Q-diversity Pruning

---

1: **procedure** THRESHOLD-BASED-PRUNING($NegSampSet, Q, \Theta, EnsSet$)
2:     **Input**: $NegSampSet$: negative samples; $Q$ the diversity metric; $\Theta$: the diversity threshold calculation function; $EnsSet$: the set of ensemble teams to be considered;
3:     **Output**: $GEnsSet$: the set of good ensemble teams.
4:     Initialize $GEnsSet = \{\}, D = \{\}$
5:     **for** $i = 1$ to $|EnsSet|$ **do**
6:         ▷ calculate the diversity metric $Q$ for $T_i \in EnsSet$
7:         $q_i = DiversityMetric(Q, T_i, NegSampSet)$
8:         $D$.append($q_i$)                      ▷ Store $q_i$ in the diversity measures $D$
9:     **end for**
10:    $\theta(Q) = \Theta(D)$                        ▷ Calculate the diversity threshold
11:    **for** $i = 1$ to $|EnsSet|$ **do**
12:        **if** $q_i < \theta(Q)$ **then**
13:           $GEnsSet$.add($T_i$)                 ▷ add qualified $T_i$
14:        **end if**
15:    **end for**
16:    **return** $GEnsSet$
17: **end procedure**

---

---

**Algorithm 2** HQ Diversity Metric Calculation

---

1: **procedure** GETHQ($NegSampSet, Q, EnsSet$)
2:     **Input**: $NegSampSet$: negative samples for each model; $Q$ the diversity metric; $EnsSet$: the set of ensemble teams to be considered;
3:     **Output**: $HQ$: the set of HQ diversity measurements
4:     Initialize $D(Q) = \{\}, \overline{D}(Q) = \{\}$
5:     Initialize $HQ = \{\}$                ▷ A map of HQ diversity metrics and teams
6:     **for** $S = 2$ to $M - 1$ **do**
7:        **for** $focal = 0$ to $M - 1$ **do**
8:           Obtain $EnsSet(F_{focal}, S)$ with candidate teams of size $S$ and containing $F_{focal}$.
9:           Initialize $D(Q, S, F_{focal}) = [\,]$
10:          **for** $i = 1$ to $|EnsSet(F_{focal}, S)|$ **do**
11:             ▷ calculate the diversity metric $Q$ for $T_i \in EnsSet(F_{focal}, S)$
12:             $q_i = DiversityMetric(Q, T_i, NegSampSet(F_{focal}))$
13:             $D(Q, S, F_{focal})$.append($q_i$)       ▷ add the $acc_i$ into $D(Q, S, F_{focal})$
14:          **end for**
15:          **for** $i = 1$ to $|EnsSet(F_{focal}, S)|$ **do**
16:             ▷ scale the diversity metrics for ensemble teams of the same size
17:             $\overline{D}(Q, S, F_{focal}, T_i) = \frac{q_i - \min(D(Q,S,F_{focal}))}{\max(D(Q,S,F_{focal})) - \min(D(Q,S,F_{focal}))}$     ▷ Scale to $[0, 1]$
18:          **end for**
19:        **end for**
20:        Obtain $EnsSet(S)$ with candidate teams of size $S$
21:        **for** $i = 1$ to $|EnsSet(S)|$ **do**
22:           Initialize $tmpD = \{\}$
23:           **for** $j = 0$ to $|T_i|$ **do**
24:             $tmpD$.append($\overline{D}(Q, S, F_{focal} = T_i[j], T_i)$)
25:           **end for**
26:           w = MemberModelAccuracyRank($T_i$)      ▷ Obtain the weights for combining $tmpD$
27:           $HQ(T_i) = WeightedAverage(w, tmpD)$
28:        **end for**
29:     **end for**
30:     **return** $HQ$
31: **end procedure**

---

HQ score. The weight is calculated with the corresponding rank of accuracy of the member model ($T_i[j]$) in the ensemble ($T_i$), i.e., the member model with higher accuracy will have higher weight (Line 21∼28).

## E    THE ALGORITHM FOR THE $\alpha$ FILTER

To construct a deep ensemble teams of diverse models, we start with building the ensembles of a smaller size, such as $S = 2$ with $\binom{M}{2} = M(M-1)$ candidates. For a larger size, such as $S^* = S+1$, we then extend these candidate ensembles of size $S$ by adding another member model from the base model pool. This way of constructing deep ensembles enables us to efficiently form high quality deep ensembles step by step and strategically prune out low diversity ensembles.

Intuitively, an ensemble team of a larger size $S = 3$, such as $[F_5, F_6, F_7]$ containing a subset of models with lower ensemble diversity (i.e., higher correlation), such as $[F_5, F_6]$, then the other teams with size $S = 3$, such as $[F_5, F_7, F_9]$, which may have higher diversity than $[F_5, F_6, F_7]$, so we can preemptively prune out $[F_5, F_6]$ for $S = 2$ to avoid calculating the diversity scores for ensembles with $S > 2$ containing $[F_5, F_6]$ as Figure 5 shows.

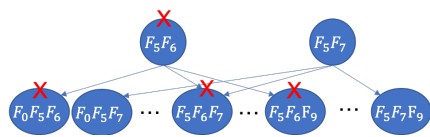

Figure 5: $\alpha$ Filter

---

**Algorithm 3** $\alpha$ Filter

---

1: **procedure** $\alpha$-FILTER($NegSampSet, Q, \beta, EnsSet$)
2:    **Input**: $NegSampSet$: negative samples; $Q$ the diversity metric; $\beta$: the percentage of number of ensemble teams to be pruned out each step; $EnsSet$: the set of ensemble teams to be considered;
3:    **Output**: $GEnsSet(Q)$: the set of good ensemble teams pruned by $\alpha$-pruning with diversity metric $Q$.
4:    Initialize $GEnsSet(Q) = \{\}$, $D = \{\}$
5:    Initialize $pruneSet = \{\}$                                                    ▷ To prune out.
6:    **for** $S = 2$ to $M$ **do**
7:        Initialize $tmpGEnsSet(Q, S) = \{\}$.
8:        Construct ensembles from $EnsSet$ of size $S$ into $EnsSet(S)$
9:        **for** $i = 1$ to $|EnsSet(S)|$ **do**
10:           **if** $T_i$ contains any group of models in pruneSet **then**
11:               continue                                                    ▷ Prune out this branch
12:           **else**
13:               $q_i = DiversityMetric(Q, T_i, NegSampSet)$
14:               $D$.append($q_i$)
15:               $tmpGEnsSet(Q, S)$.append($T_i$)
16:           **end if**
17:        **end for**
18:        $n = \beta * |tmpGEnsSet(Q, S)|$
19:        sort $T_i \in tmpGEnsSet(Q, S)$ by $q_i$
20:        remove $n$ teams of lowest diversity from $tmpGEnsSet(Q, S)$ and add them in to $pruneSet$
21:        $GEnsSet(Q) \cup = tmpGEnsSet(Q, S)$
22:    **end for**
23:    **return** $GEnsSet(Q)$
24: **end procedure**

---

Therefore, with this property we can effectively prune out low-diversity deep ensembles. Algorithm 3 presents a skeleton of the pseudo code, describing this pruning process. $NegSampSet$ contains the set of negative samples for calculating the diversity metric $Q$. $\beta$ marks the percentage of the teams to be further pruned out for a fixed team size. By default, we set $\beta = 10\%$. $EnsSet$ contains the set of ensemble teams to be considered. For each team size, we omit all the teams that contain any group of models in $pruneSet$. For the remaining teams, we measure their diversity scores and ordered them based on the diversity score $p_i$. Then we remove $\beta$ of the remaining teams with the lowest diversity and add them into $pruneSet$ for further pruning. This algorithm can significantly avoid exploring unpromising branches in searching for high-quality ensembles.

## F  THE BASE MODEL POOLS FOR THREE BENCHMARK DATASETS

We evaluate the proposed hierarchical diversity pruning methods using three benchmark datasets, CIFAR-10, ImageNet, and Cora. The specification of these datasets and the base model pools for each of the datasets are included in this section as Table 5 shows. We use 10 base models in this study for each dataset, primarily collected from GTModelZoo (GTModelZoo Developers, 2020).

Table 5: Base Model Pools for Three Benchmark Datasets

| Dataset | CIFAR-10 | | ImageNet | | Cora | |
|---|---|---|---|---|---|---|
| | 10,000 testing samples | | 50,000 testing samples | | 1,000 testing samples | |
| Number | Models | Accuracy (%) | Models | Accuracy (%) | Models | Accuracy (%) |
| 0 | DenseNet190 | 96.68 | AlexNet | 56.63 | GCN | 81.70 |
| 1 | DenseNet100 | 95.46 | DenseNet | 77.15 | GAT | 82.80 |
| 2 | ResNeXt | 96.23 | EfficientNet-B0 | 75.80 | SGC | 81.70 |
| 3 | WRN | 96.21 | ResNeXt50 | 77.40 | ARMA | 82.10 |
| 4 | VGG19 | 93.34 | Inception3 | 77.25 | APPNP | 82.20 |
| 5 | ResNet20 | 91.73 | ResNet152 | 78.25 | APPNP1 | 83.80 |
| 6 | ResNet32 | 92.63 | ResNet18 | 69.64 | APPNP2 | 88.70 |
| 7 | ResNet44 | 93.10 | SqueezeNet | 58.00 | SplineCNN | 88.90 |
| 8 | ResNet56 | 93.39 | VGG16 | 71.63 | SplineCNN1 | 88.30 |
| 9 | ResNet110 | 93.68 | VGG19-BN | 74.22 | SplineCNN2 | 88.50 |
| MIN | ResNet20 | 91.73 | AlexNet | 56.63 | GCN/SGC | 81.70 |
| AVG | | 94.25 | | 71.60 | | 84.87 |
| MAX | DenseNet190 | 96.68 | ResNet152 | 78.25 | SplineCNN | 88.90 |

Table 6: Q-Metrics with $\alpha$ filter

| Dataset | Methods | #EnsSet | #GEnsSet | Ensemble Acc Range (%) | #(%)(Acc > max) |
|---|---|---|---|---|---|
| CIFAR-10 | Q-CK | 270 | 242 | 93.56~96.64 | 0 |
| | Q-QS | 269 | 242 | 93.56~97.01 | 4 (1.65%) |
| | Q-BD | 241 | 241 | 93.56~96.64 | 0 |
| | Q-FK | 266 | 238 | 93.56~96.64 | 0 |
| | Q-KW | 268 | 241 | 93.56~96.64 | 0 |
| | Q-GD | 236 | 213 | 93.56~97.01 | 16 (7.51%) |
| ImageNet | Q-CK | 260 | 235 | 61.39~80.01 | 56 (26.83%) |
| | Q-QS | 253 | 229 | 61.39~80.01 | 64 (27.95%) |
| | Q-BD | 277 | 249 | 61.39~79.84 | 67 (26.91%) |
| | Q-FK | 260 | 235 | 61.39~80.01 | 56 (23.83%) |
| | Q-KW | 277 | 249 | 61.39~79.84 | 67 (26.91%) |
| | Q-GD | 237 | 215 | 68.99~80.29 | 104 (48.37%) |
| Cora | Q-CK | 211 | 185 | 82.10~89.50 | 9 (4.86) |
| | Q-QS | 214 | 187 | 82.10~89.50 | 9 (4.81%) |
| | Q-BD | 215 | 188 | 82.10~89.50 | 9 (4.79%) |
| | Q-FK | 211 | 185 | 82.10~89.50 | 9 (4.86%) |
| | Q-KW | 215 | 188 | 82.10~89.50 | 9 (4.79%) |
| | Q-GD | 228 | 208 | 82.10~89.50 | 12 (5.77%) |

## G  THE $\alpha$ FILTER ON Q DIVERSITY METRICS

We also applied the $\alpha$ filter with six Q-diversity metrics for pruning out low-diversity ensembles. Figure 6 shows the experimental results on the Q-GD metric on CIFAR-10, where Figure 6a, 6b and 6c present all the candidate ensembles of size 3, 4 and 5 respectively, and the relationship of the Q-diversity metric GD and ensemble accuracy. The black dots mark the ensembles that pruned out by the $\alpha$ filter while the red ones represent the remaining ensembles. Even though, the $\alpha$ filter can significantly filter many low-diversity ensembles, we still miss a fair number of ensembles with high ensemble accuracy. There are two primary reasons behind this observation: (1) The Q-diversity metrics fail to precisely capture the diversity of ensembles, therefore, when pruning out a low Q-diversity branch, such as in Figure 6a with $S = 3$, some ensembles of a larger size with high diversity (with low Q-GD values) many also be pruned out in Figure 6b with $S = 4$. (2) A few ensembles

with high ensemble accuracy have low diversity, demonstrating that Q-diversity metrics may not be effectively correlated to ensemble accuracy. We further perform a comprehensive evaluation as Table 6 shows on three datasets. Due to the above inherent problems with Q metrics, the $\alpha$ filter on our HQ metrics achieved much better performance than Q metrics, when comparing Table 6 with Table 2, 3, 7.

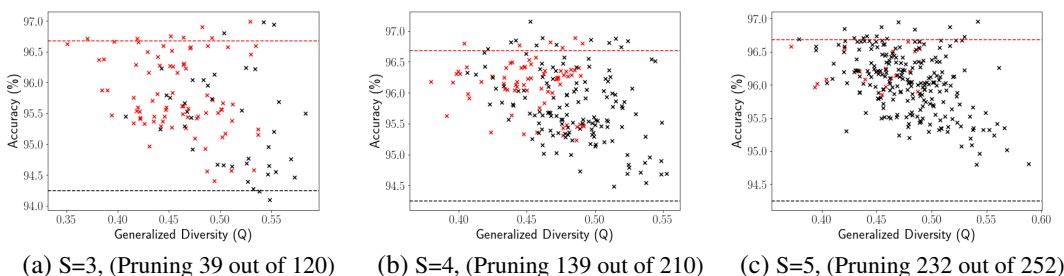

(a) S=3, (Pruning 39 out of 120)    (b) S=4, (Pruning 139 out of 210)    (c) S=5, (Pruning 232 out of 252)

Figure 6: $\alpha$ filter on Q diversity with different team size $S$ (CIFAR-10, **Q- GD**))

## H EXPERIMENTAL EVALUATION ON CORA DATASET

We also evaluate our methods on a popular graph dataset, Cora. The same set of experimental results are shown on Table 7. We found similar observations as CIFAR-10 and ImageNet. *First*, the $\alpha$ filter with HQ metrics works much better than Q metrics. HQ metrics can capture more high accuracy ($\geq 89\%$) ensembles (14~18) than 6~17 by Q metrics with the mean threshold. *Second*, the combined hierarchical pruning method of the $\alpha$ filter and K-Means filter on HQ metrics ($\alpha + K$) can significantly improve the ensemble accuracy lower bound from 82.10% to 86.70%~87.80% as well as the probability of high accuracy ensembles among the selected ones.

Table 7: Comparing Q, HQ ($\alpha$), and HQ ($\alpha + K$) metrics on Cora

| Methods | #EnsSet | #GEnsSet | Ensemble Acc Range (%) | #(%)(Acc > 88.90% (max)) | #(Acc >= 89.00%) |
|---|---|---|---|---|---|
| Baseline | 1013 | 1013 | 82.10~89.50 | 38 (3.75%) | 22 |
| Q-CK | 1013 | 594 | 85.00~89.50 | 21 (3.54%) | 12 |
| Q-QS | 1013 | 596 | 85.40~89.50 | 23 (3.86%) | 14 |
| Q-BD | 1013 | 602 | 85.00~89.50 | 22 (3.65%) | 13 |
| Q-FK | 1013 | 583 | 85.00~89.50 | 21 (3.60%) | 12 |
| Q-KW | 1013 | 647 | 86.10~89.10 | 13 (2.10%) | 6 |
| Q-GD | 1013 | 583 | 85.40~89.50 | 32 (5.49%) | 17 |
| HQ-CK ($\alpha$) | 234 | 212 | 82.10~89.50 | 21 (9.91%) | 15 |
| HQ-QS ($\alpha$) | 256 | 232 | 82.50~89.50 | 24 (10.34%) | 18 |
| HQ-BD ($\alpha$) | 241 | 218 | 82.10~89.50 | 20 (9.17%) | 14 |
| HQ-FK ($\alpha$) | 234 | 212 | 82.10~89.50 | 21 (9.91%) | 15 |
| HQ-KW ($\alpha$) | 241 | 218 | 82.10~89.50 | 20 (9.17%) | 14 |
| HQ-GD ($\alpha$) | 239 | 216 | 82.10~89.50 | 20 (9.26%) | 14 |
| HQ-CK ($\alpha + K$) | 212 | 22 | 86.70~89.50 | 5 (22.73%) | 4 |
| HQ-QS ($\alpha + K$) | 232 | 67 | 87.80~89.00 | 8 (11.94%) | 5 |
| HQ-BD ($\alpha + K$) | 218 | 26 | 86.70~89.50 | 5 (19.23%) | 4 |
| HQ-FK ($\alpha + K$) | 212 | 24 | 86.70~89.50 | 5 (20.83%) | 4 |
| HQ-KW ($\alpha + K$) | 218 | 30 | 86.70~89.50 | 4 (13.33%) | 3 |
| HQ-GD ($\alpha + K$) | 216 | 24 | 86.70~89.50 | 5 (20.83%) | 4 |

