# OpenReview forum: "Deep Ensembles with Hierarchical Diversity Pruning"
_ICLR.cc/2021/Conference — Reject_

### Official Review · AnonReviewer2 · 2020-10-27
**This study proposes an experimental analysis of ensemble diversity metrics also a heuristic filtering strategy for not diverse model pruning. In the proposed strategy,  ensembles (with a small fixed size) are teamed up and the ones with high HQ scores and their super-sets are pruned out. This stage is followed by a (K=2)-means clustering stage, in which the ensembles are clustered to low-HQ, high accuracy or low accuracy, high-HQ, to prune out those ensembles with low accuracy and low diversity.**

**Rating:** 4
**Confidence:** 4

**Review:**

The paper is well-written, and extensive experimental results are presented. However, I think this study lacks theoretical supports to back the statements that have been obtained experimentally.
Also, the contributions made by the paper do not seem significant.
HQ-diversity metrics proposed in this study, as the authors also mention it, are an extension of the Q-diversity metrics (first paragraph of section 2.1).
In the proposed extension, HQ-diversity metrics are a normalized (scaled) version of the Q-diversity metrics in which the model accuracy rank is also considered. One question that arises here is whether the obtained improvement in accuracy is because of the normalization step or considering the accuracy rank to weight the models or both steps are essential. Also, I am wondering how costly would be validating the accuracy of the models. In my opinion, it will be more reasonable to include a time complexity comparison too.
Regarding the presented strategy for pruning, I am not sure how efficient is the proposed filtering strategy in a real application when one needs to generate many ensembles of different sizes and then go through a costly stage-wise pruning.

---

### Official Review · AnonReviewer4 · 2020-10-28
**Recommendation to Reject**

**Rating:** 4
**Confidence:** 4

**Review:**

Summary:
The manuscript studies the problem of ensemble selection (pruning) with the ensemble consists of deep neural network models. The authors compare different diversity metrics, which they named collectively as Q-metric, visualize the accuracies of different ensembles on CIFAR-10 dataset where the ensembles are stratified by their sizes. Based on their observation,  the authors further propose HQ-metric, HQ(\alpha) and HQ(\alpha +K) to improve the diversity score from Q-metrics. The authors evaluate their strategies on CIFAR-10 and  on all of the Q-metircs  and show that those Q-metric, when incorporating their proposed strategies, in general, is capable of selecting ensembles of higher accuracy.

Strengths:

1 The manuscript provides somewhat detailed analysis of correlation between diversity score obtained from KW, GD and ensemble accuracy with the ensembles stratified by their respective sizes.
2 The manuscript attempts in different ways to improve the diversity score from those metrics and empirically evaluated their strategies on ensemble models train on CIFAR-10 and ImageNet.

Weaknesses

1.	Biases in the analysis

On page 3, it is mentioned that “ Given a Q-diversity metric, we calculate the diversity score for each ensemble team in the ensemble set (EnsSet) using a set of negative samples (NegSampSet)” .
Why is it that only negative samples are used in the analysis? In this case, is it fair to say that the results from the analysis is biased (since the distribution of the testing set is different from training set)?

2	Unsubstantiated claims/arguments

a)	On page 4  “The Qdiversity metrics may not accurately capture the degree of negative correlation among the member models of an ensemble even when its ensemble Q-diversity score is below the mean threshold. ”
              Can authors verify this? Can authors actually show how inaccurate the Q-metrics are in measuring the negative correlation between ensemble members?
b)	On page 5 “These two optimizations enable HQ scores to more accurately capture the ensemble diversity and its relationship with the ensemble accuracy”
Why is it so? Can authors provide a more rigorous proof to this, instead of just showing some empirical examples?

3.	Confusing to read

a)	On page 4, it is mentioned that “Comparing ensembles of different team size S using their Q scores may not be a fair measure of their true ensemble diversity in terms of the degree of negative correlation among member models of an ensemble”
Can authors explain what is exactly “true ensemble diversity in terms of the degree of negative correlation among member models”? Aren’t  ensemble diversity and negative correlation two different concepts? Why negative correlation matters here? Why not non-correlatedness? The authors have mentioned negative correlation multiple times as the “limitations” of previous approaches, giving readers an impression that it is a crucial property for a good ensemble of DNN models to have and yet offering no evidence or proof that ensembles with high negative correlation are more likely to yield better accuracy. Finally, the authors did not show that the ensembles selected by their approach do, on average, have higher negative correlation among the ensemble members

b)	On page 5 “we argue that comparing ensembles of the same team size in terms of their diversity scores can better capture the intrinsic relationship between ensemble diversity and ensemble accuracy”
Can authors give a more formal definition to “the intrinsic relationship between ensemble diversity and ensemble accuracy” and at same time give some explanation as to why comparing the diversity scores of the ensembles of the same size can better capture such relationship?

4.	 Way too empirical

Overall, the strategies proposed in the manuscript rely heavily on the empirical observations from the performance of a few models trained on one dataset. Thus, it is hard to say how applicable such strategies are when the scenario slight changes (say, the number of base models for selection is at the magnitude of hundreds or thousands or even more, or the dataset for training models is a much smaller dataset from a specialized domain).
For instance, on p.5 “We observe that if an ensemble team of size S has large HQ score (say [F5, F6]), indicating insufficient ensemble diversity, then all the ensemble teams that have larger size than S and contain all the member models of this ensemble team (e.g., [F5, F6, F7], [F0, F5, F6], [F0, F5, F6, F7], [F5, F6, F7, F8]) tend to have insufficient ensemble diversity”. Yet Fig.1(a) does show that the ensemble of a larger size (colored red) have lower scores. So the question becomes is it possible that the effects from the errors made in the early stages could be undone by the correct choices in later steps, particularly when there is a large pool of base models?   Since the ensemble selection problem is essentially a combinatorial optimization problem, there might exist a more principled approach to this.  See the following reference as an example.
Bühlmann, Peter, and Nicolai Meinshausen. "Magging: maximin aggregation for inhomogeneous large-scale data." Proceedings of the IEEE 104.1 (2015): 126-135.

5.	Some mistake

On page 5  should “(M,  2)  = M(M − 1) candidate ensemble” actually be “M(M − 1)/2” . So should  “ For M = 10 we will have 90 teams of size 2” be “45 teams of size 2”

---

### Official Review · AnonReviewer1 · 2020-10-29
**This paper identifies variations on the ensemble diversity metrics that correlate better with accuracy of the ensemble and therefore can be used to select the base models that should be used in the ensemble. The paper demonstrates experimentally that their metrics do correlate better with accuracy and so they are able to select better ensembles.**

**Rating:** 3
**Confidence:** 4

**Review:**

The paper succeeds in developing diversity metrics that correlate better with ensemble accuracy than the original diversity metrics. However, this makes one wonder why one cannot just use ensemble accuracy directly. One can also use a combining scheme along the lines of (Freund, 1995) where it adds models that focus on the examples that will increase accuracy and allowing errors on examples where most of the models so far have either classified the examples correctly already or incorrectly (where there is no hope of recovery and so effort is not worthwhile). Additionally, the appendix has the algorithms and other substantive content that is central to the paper, which is not supposed to be the case.

Here are additional comments.

1. Bagging does not necessarily (or typically) aim to train weak models to form a strong ensemble. In fact, by default, it wants the base models to be much more accurate than random, but with large variance among them.
2. The Random Forest reference is incorrect. Please use Breiman's Machine Learning Journal paper from 2001 as the reference since that is the original one.
3. In appendix B, note that $\delta = 0$ does not mean that the base models are negatively correlated, but rather they are uncorrelated. For negative correlation, ideally, $\delta = \frac{-1}{S-1}$ since that would lead to $E_{avg}^{add} = 0$.

---

### Official Review · AnonReviewer3 · 2020-10-29
**This paper attempts to introduce new diversity measures for ensembles that better correlates with accuracy, thus enabling an effective selection strategy of the components.**

**Rating:** 3
**Confidence:** 5

**Review:**

The major issue with this paper is that the definitions of the new diversity measures HQ are not properly laid out, and an explanation/intuition why they would correlate with accuracy is not provided. I understand that the algorithm is provided in the Appendix, but it comes with barely any explanation still. The explanations and motivation in 2.2. are vague, contain undefined terms and concepts, and they are completely unclear. The idea of focal model (which seems to be crucial) is never explained. Furthermore, it's unclear why this paper focuses on deep learning models. Do the proposed diversity measures work only for deep learning models? Why? Would they work also on other models, and regardless of the models used for the ensemble components? What is about these measures that make them correlate with accuracy?

- The authors should clarify early on in the paper that the focus of this work is on ensembles for supervised learning. Work has been done also on selective ensemble techniques for unsupervised learning (e.g. clustering or anomaly detection), but this is not the scope of this work.

- I find the discussion in Section 2 quite lengthy and trivial. It's well known that enumerating all subsets of an ensemble of a given size is unfeasible. I would shorten it to leave space to core explanations.

- Results of Table 1: the authors state that the mean threshold is computed on 1013 candidate deep ensembles. Which deep models were used? The authors need also to clarify that M=10 (this becomes clear only later) in this experiment, and how the individual subsets were generated (randomly?). State upfront what's the purpose of this experiment.

- Figure 1: I don't think (c) is showing much of "a sharper trend in terms of the relationship between ensemble diversity and ensemble accuracy" as claimed by the authors. For a given diversity value below the threshold, there is still a range of accuracies being obtained by the ensembles with that diversity level. So I am not sure what's the added value of plot (c). I understand the authors are trying to motivate their choice of comparing ensembles of the same size, but this plot does not serve the purpose well.

- When introducing the concept of focal model in 2.2: "..., we introduce the concept of focal model to obtain the set of negative samples for computing the diversity scores of ensembles by taking each member model in turn as the focal model." This statement is totally obscure to me. What are the negative samples? How is a focal model selected. The explanations in this paragraph are not useful.

- The HQ (alpha) method does not work that well. It's comparable to standard diversity measures on CIFAR-10, and worse than standard diversity measures on Cora. This raises the question whether it's basically the extra step of clustering that gives an advantage to HQ (alpha + K).

---

### Decision · Program_Chairs · 2021-01-07
**Final Decision**

**Decision:**

Reject

**Comment:**

This work studies statistics of ensemble models that capture the prediction diversity between ensemble members.  The goal of the work is to identify or construct a metric which is predictive of the holdout accuracy achieved by the ensemble prediction.

Pros:
* Studies empirically how measures of ensemble diversity relate to ensemble prediction accuracy.
* Proposes improvements to diversity metrics that correlate better with accuracy.

Cons:
* Unclear/confusing presentation.
* Limited empirical validation that relies mostly on CIFAR-10 results to justify claims
* Some claims made (trend between ensemble diversity and accuracy, Q diversity capturing not capturing negative correlations) are not substantiated.

All reviewers recommend this paper to be rejected and the authors did not reply to any reviews.